# The Multifaceted Role of Jasmonic Acid in Plant Stress Mitigation: An Overview

**DOI:** 10.3390/plants12233982

**Published:** 2023-11-27

**Authors:** Muhammad Rehman, Muhammad Sulaman Saeed, Xingming Fan, Abdul Salam, Raheel Munir, Muhammad Umair Yasin, Ali Raza Khan, Sajid Muhammad, Bahar Ali, Imran Ali, Jamshaid Khan, Yinbo Gan

**Affiliations:** 1Zhejiang Key Lab of Crop Germplasm, Department of Agronomy, College of Agriculture and Biotechnology, Zhejiang University, Hangzhou 310058, China; m.rehmanafridi789@gmail.com (M.R.);; 2Institute of Food Crops, Yunnan Academy of Agricultural Sciences, Kunming 650204, China; 3Department of Botany, Kohat University Science and Technology, Kohat 26000, Pakistan; 4Department of Biotechnology and Genetic Engineering, Kohat University Science and Technology, Kohat 26000, Pakistan

**Keywords:** abiotic stresses, biosynthesis, jasmonic acid, Jasmonates, plant hormones, plant growth, signaling

## Abstract

Plants, being sessile, have developed complex signaling and response mechanisms to cope with biotic and abiotic stressors. Recent investigations have revealed the significant contribution of phytohormones in enabling plants to endure unfavorable conditions. Among these phytohormones, jasmonic acid (JA) and its derivatives, collectively referred to as jasmonates (JAs), are of particular importance and are involved in diverse signal transduction pathways to regulate various physiological and molecular processes in plants, thus protecting plants from the lethal impacts of abiotic and biotic stressors. Jasmonic acid has emerged as a central player in plant defense against biotic stress and in alleviating multiple abiotic stressors in plants, such as drought, salinity, vernalization, and heavy metal exposure. Furthermore, as a growth regulator, JA operates in conjunction with other phytohormones through a complex signaling cascade to balance plant growth and development against stresses. Although studies have reported the intricate nature of JA as a biomolecular entity for the mitigation of abiotic stressors, their underlying mechanism and biosynthetic pathways remain poorly understood. Therefore, this review offers an overview of recent progress made in understanding the biosynthesis of JA, elucidates the complexities of its signal transduction pathways, and emphasizes its pivotal role in mitigating abiotic and biotic stressors. Moreover, we also discuss current issues and future research directions for JAs in plant stress responses.

## 1. Introduction

Plant growth and development are profoundly affected by both biotic and abiotic stress factors. The abiotic factors include drought, salinity, water logging, heat, freezing, and heavy metal stress [1,2,3,4]. Drought stress causes osmotic stress which results in dehydration of plant cells leading to plant death [4]. Salinity is also very lethal for the osmoregulation system in plants [3]. Waterlogging stressors, commonly known as flooding stress, affect the species distribution and growth of many different communities of plants [5]. Heat and cold stresses negatively affect plant development. Among these, heavy metal toxicity is of more serious concern since it affects plant growth and development resulting in food safety and food security. Heavy metals refer to metals and metalloids that have higher densities, typically more than 5 g/cm^3^ [6]. These metals, when taken and absorbed by the plant root, cause many problems for plants, such as disturbed redox homeostasis, imbalanced electrolytes, electron transport chains, etc. [7]. Global climatic conditions are continuously changing, posing a serious threat to plant and human health. Therefore, it is necessary to compete with the changing climatic conditions in order to overcome environmental challenges and safeguard food security as well as food safety [8,9,10]. Plant growth and development are generally dependent on metabolic pathways, including REDOX processes, which allow them to synthesize their food in the form of ATPs [11]. However, when heavy metals (copper, arsenic, zinc, cobalt, chromium, manganese, etc.) are present in the soil, they ultimately disrupt the plant metabolism which results in an inhibition of plant growth [12,13,14,15,16,17]. These heavy metals are highly insoluble and remain in the soil for long periods. In the process of water uptake, heavy metals enter via root cells into the xylem, hindering plant metabolic pathways, and ultimately leading to plant cell death [18]. Phytohormones are known to regulate plant growth and development and induce stress tolerance against the aforementioned stresses. Among them, jasmonic acid (JA) is one of the vital hormones that has a role in plant signaling pathways which can induce biochemical and physiological modifications and helps plants mitigate the lethal effects of various abiotic stresses [19]. Many studies have been conducted on the specific role of exogenous and endogenous JA against abiotic stresses in plants. Nevertheless, understanding the complex processes that regulate the synthesis and function of JA in plants is a highly challenging process characterized by numerous intricate stages and molecular pathways. The activation of diverse gene actions intricately regulates the generation of JA. However, our understanding of the underlying pathways involved in this process remains somewhat shallow. Nonetheless, it is imperative to acquire a comprehensive knowledge of JA, particularly its role in inducing stress tolerance. The primary objectives of this review center on achieving an exhaustive grasp of JA production, its mode of action, and its practical applications. Within the scope of this review, we will expand upon the detailed biosynthesis, pathways, and the role of JA in counteracting various abiotic stress conditions.

## 2. Plant Hormones Play Significant Roles in Stress Alleviation

Phytohormones, also known as plant hormones, are intrinsic chemical compounds that govern various physiological and biochemical plant processes when present in minimal concentrations. There are nine different classes of phytohormones reported [20,21,22]. According to their structural and chemical diversity, they are classified into auxin, cytokinins (CKs), ethylene, gibberellins (GAs), abscisic acid (ABA), salicylic acid (SA), brassinosteroids (BRs), strigolactones (SLs), and jasmonic acid [23]. Phytohormones play a very crucial role in regulating normal plant growth and maintaining the homeostasis of plant cells against abiotic stress. They are bio-stimulators and signaling molecules that induce the exocytosis process of cells under adverse conditions [24]. In addition to increasing plants’ resistance against abiotic stresses, like salinity and drought, phytohormones also perform vital regulatory roles against hazardous compounds [25,26]. Phytohormones protect plants against cellular-oxidative stress by increasing photosynthesis, and total chlorophyll content, as well as plant development and growth while reducing the production of reactive oxygen species [27]. Ionic toxicity is a direct consequence arising from the excessive accumulation of Na^+^ in plants subjected to saline conditions. The physiological integrity of plant cells relies on the particular maintenance of intracellular ionic equilibrium, a balance that can be disrupted by such conditions [28]. To uphold ionic homeostasis, jasmonic Acids (JAs) play a pivotal role. Notably, the introduction of JA to maize seedlings experiencing Na_2_CO_3_-induced stress has been observed to mitigate Na^+^/K^+^ ratios in both roots and leaves. This mitigation, in turn, alleviates ionic toxicity and reduces the harmful effects associated with alkaline stress [29]. Exogenous methyl jasmonate (MeJA) emerges as a crucial factor in preserving ion homeostasis in seedlings treated with NaCl, evidenced by its capacity to reduce Na+ accumulation and modulate the distribution of K^+^ between roots and shoots. This modulation results in a substantial decrease in the shoot Na^+^/K^+^ ratio. The accumulation of K^+^ in the shoots of salt-stressed plants enables them to sustain vital processes such as photosynthesis, metabolism, and osmotic pressure, concurrently mitigating ion toxicity [30]. These hormones induce different responses according to their mode of action. Here in this review, we discuss the recent advances in biosynthesis and signaling pathways of JA and their role in mitigating abiotic and biotic stresses in detail.

## 3. Jasmonic Acid as a Plant Biostimulant

Jasmonic acid is an endogenous biomolecule in plants that serves as a regulator of plant growth, initially characterized as a hormone associated with stress responses in higher plant species. They were first discovered in plants in 1962 and were identified to play a role in plant biological systems such as stress tolerance, movement, and reproductive growth of the plant cells [31,32,33]. These are carboxylic acids and hormonal lipids in nature, belonging to the class of oxylipins. Jasmonic acid and its derivatives are overall called jasmonates (JAs). Some of these derivatives include *cis-jasmone*, *methyl jasmonate*, *jasmonate-amino acid*, lactones of *12-hydroxy-JA-IIe*, *12-O-glucosyl-JA-IIe*, and *jasmonic Isoleucine*, etc. [34,35,36,37]. They are derived from the cyclopentanones compounds which link to the family of oxylipins. Therefore, these molecules are categorized as a family of lipids that play a vital role in cell signaling against abiotic stresses such as drought, salinity, heavy metals, etc. The concentration of JA varies in plant tissue. For example, in flowers and reproductive parts, it is very high, whereas its concentration is low in mature roots and leaves [11,38]. The role of JA is very crucial since it has multiple functions in plant cells like cell division, reproductive parts growth, fruit ripening, phosphorous and nitrogen uptake, electron transport chain, stomatal opening, and glucose transport [39]. Moreover, jasmonic acid plays multifaceted roles in various physiological processes, including fruit ripening, pollen survival, and the development of roots and stems. It is also integral to the plant response to damage and acts as a key component in biotic stress responses, conferring resistance to pathogens and insects [40,41]. In the context of climacteric fruits, such as apples, the application of MeJA has been shown to enhance the development of desirable characteristics. Specifically, MeJA application enhances the red coloration, anthocyanin, and β-carotene contents, along with elevating concentrations of numerous phenolic components. Additionally, MeJA induces the accumulation of ethylene and various ester compounds in climacteric fruits like apples [42,43]. Conversely, non-climacteric fruits such as strawberries, blackberries, and raspberries exhibit distinct responses to MeJA treatment. Following MeJA application, there is an observed increase in the concentrations of sucrose and glucose, the soluble solid content/titratable acidity (SSC/TA) ratio, and the anthocyanin content in these fruits [44,45,46]. JA functions as a signaling molecule that modulates the expression of stress-responsive genes in response to abiotic stress conditions [47,48]. It is observed, by conducting qRT-PCR studies, that JA has a diverse strategy to induce tolerance against heavy metal lead (Pb) in tomato plants [48,49,50]. Another study was carried out on *Alyssum inflatum Náyr* populations by applying nickel metal particles on the plant to investigate the harmful effects of Pb and showed that JA resisted completely against Nickel stress. Studies have also shown that *Avicennia marina* plant seedlings showed resistance against cadmium and copper by exogenously applying JA, inducing the stimulatory effects [37,48,51].

In conclusion, JA has the ability to induce tolerance against heavy metals like lead (Pb), nickel, cadmium, and copper in different plant species, underscoring its potential for use in phytoremediation and stress management strategies. However, the molecular mechanisms through which JA exerts its effects on plant cells are poorly understood. A more in-depth discussion of the signaling pathways and target genes involved in JA-mediated stress responses would enhance the scientific understanding of this hormone.

## 4. Biosynthesis of Jasmonic Acid

Jasmonic acid is an essential hormone that is found in plant cells. It is involved in the responding capability of plant cells against stress [18,49,52]. For the first time, JA was obtained from the *Jasminum grandiflorum* in the form of methyl ester JA. JA is categorized as a cyclopentane fatty acid. It is produced from *α-linolenic acid*, an important part of different membranes of plant cells (Figure 1) [53,54,55]. 

Moreover, jasmonic acid (JA) is synthesized within cellular compartments through enzymatic catalysis mediated by enzymes such as lipoxygenase and allene oxide cyclase, which govern the regulation of octadecanoid metabolic pathways. The process of signaling and releasing JA takes place in the plastids where α-linolenic acid is oxygenized into *13S-hydroperoxyoctadecatrienoic acid* by the catalysis of *LOX*, then *13-HPOT* is converted into the form of *cis-(+)-12-oxophytodienoic acid* by the activity of *AOC* and *AOS*. Then, *OPDA* is translocated to peroxisomes where *OPDA* is completely reduced by *OPR* (Figure 1). Subsequently, it is reduced to a form of *(+)-7-iso-JA* in three cycles of β-oxidation. Then, *(+)-7-iso-JA* is shifted towards the cytoplasm where, with the catalysis of *jasmonate-amido synthetase* and *jasmonate methyl transferase*, it is converted into *JA-isoleucine or methyl jasmonate* [22,53,56].

The plants that are highly or slightly exposed to abiotic stress enhance the rate of production and release of JA by switching on the genes of JA. This shows that JA is a very important hormone against abiotic stress. 

In conclusion, this study offers a foundation for understanding the biosynthesis and regulatory role of JA in plant cells. However, further research is needed to address the mentioned deficiencies, including investigating the sources of JA in different plant species, elucidating the regulation of JA production, and exploring its specific biological functions in response to abiotic stress and biotic.

## 5. Signaling Transduction of Jasmonic Acid

Many investigations have been conducted to unravel the signaling pathway of JA in plants. It is evident from studies that in response to abiotic stress, there is a prompt induction of genes associated with the synthesis of JA, leading to the activation of the plant’s defense mechanism through the regulated accumulation of JA within plant cells [47,57,58,59]. There are specific stress-responsive genes that switch on upon exposure to stress. The pathway of JA release is composed of *CORONATINE INSENSITIVE 1*(COI1), a receptor, *JASMONATE ZIM-DOMAIN PROTEIN (JAZ)* repressors, and the JA transcription factor known as *MYC2*. During its expression, the *JA-Ile* intermingles with the JA receptor (COI1). This COI1 is the F-box protein that works in the *E3-ubiquitin ligase-mediated* proteolysis of the *JAZs*. For the signaling pathway, *MYC2* is known as a vital transcription factor involved in this process (Figure 2). The suppression effect of jasmonic acid genes was observed in Arabidopsis when the phenotype of the mutants of *jasmonate insensitive 1 (jin1)* was analyzed in 1996 [60].

In 2018, an experimental study showed that JA-inducing genes were suppressed when they were exposed to lead toxicity in tomato plants. Furthermore, these genes were rarely expressed and were unable to perform their respective functions under lead toxicity [61]. Further research studies revealed that the *JIN1* gene encodes the transcription factor *MYC2* which enhances the expression of JA-responsive genes by interacting with the *G-Box (5’-CACGTG-3’)* in their specific promoter parts [62,63]. These findings show that the activation of *MYC2* is crucial for the expression of JA-responding genes and also for the production of JA response. COI1-*JA-Ile* interrelation causes the induction of ubiquitin-mediated proteolysis of *JAZs*, which is further followed by the production of *MYC2* transcription factors from the *JAZ-MYC2* complexes along with the transcriptions of JA-responsive genes. In transgenic plants, the role of *MYC2* in JA signaling is very well understood which is elaborated by overexpression and knock-out systems. Moreover, many diverse kinds of *MYC2* orthologs have been reported in different crops. We could conclude that *MYC2* specifically co-regulates JA-dependent genes [9,19,31,62,64].

## 6. Role of JA during Plant Defense Responses

The JA signaling pathway involves critical components, such as COI1, JAZ repressors, MYC2 as a vital transcription factor and the role of JA-Ile in activating this pathway. However, more detailed information on how COI1, JAZs, MYC2, and other components interact would enhance this clarity. The possible action of different genes against abiotic stresses responsive to jasmonates identified in many trials is given below in Table 1.

In the dynamic and constantly changing natural environment, plants may experience a multitude of biotic factors, including a wide array of pathogens, encompassing biotrophic, hemi-biotrophic, and necrotrophic pathogenic organisms, which are detrimental to the optimal development of plants [74]. Plant resistance responses to these stresses, especially to insects, are controlled by molecular signals, among which the most significant signaling molecule is JA. Certainly, lipid-derived jasmonates (JAs) play a pivotal role in various aspects of plant development and serve as integral components in the plant’s defense mechanisms against diverse pathogens, specifically necrotrophic fungal pathogens such as *Alternaria brassicicola*, *Botrytis cinerea*, *Plectosphaerella cucumerina*, and *Pythium* spp. These pathogens pose substantial threats to plant viability and growth [75,76]. Pathogen attack stimulates the biosynthesis of jasmonoyl-L-isoleucine (JA-Ile), which subsequently engages with the COI1-JAZ receptor, instigating the degradation of JAZ repressor proteins and the commencement of transcriptional processes associated with the activation of plant defense mechanisms (Figure 3). Interestingly, certain highly virulent pathogens have evolved distinct strategies to inhibit the JA signaling pathway and facilitate their utilization of plants as host organisms [75,77].

The JA signaling pathway comprises two distinct phases: repression, which occurs under normal physiological conditions, and activation, which is triggered in response to stress conditions. During normal conditions, the cytoplasm typically maintains a low concentration of JA-Ile, thereby causing the genes associated with JA to remain in an inactive state [78]. Gene promoters exhibit an affinity for various transcription factors (TFs), which are subjected to repression by various transcriptional repressors known as JAZ proteins, a group characterized by the presence of the jasmonate ZIM domain [79]. JAZ proteins initiate the formation of an active, closed complex referred to as the JAZ–NINJA–TPL complex by recruiting the protein known as topless (TPL) and the adaptor protein specific to JAZ (NINJA). This complex avoids the start of jasmonate-responsive genes [80]. During stress, the phytohormone jasmonic Acid-Isoleucine (JA-Ile) exhibits an increased accumulation within the cellular cytosol. Subsequently, these JA-Ile molecules are transported across cellular membranes to access the nucleus, facilitated by the catalytic activity of jasmonic Acid Transfer Proteins, specifically AtJAT1/AtABCG16. This process marks the initiation of the JA signaling pathway [78]. Within the cellular nucleus, the SCF complex, composed of kinetochore protein 1 (SKP1), cullin 1 (CUL1), and an F-box protein, serves as an E3 ubiquitin ligase that plays a pivotal role in facilitating JA responses. The initial step involves the translocation of JA-Isoleucine (JA-Ile) into the nucleus, where it is recognized by the F-box protein COI1, an integral component of the SCF complex. Subsequently, the COI1-JAZ co-receptor complex is activated upon detection of JA-Ile, thereby facilitating the interaction between JAZ (jasmonate ZIM-domain) proteins and COI1. This interaction triggers the degradation of JAZ proteins within the 26S proteasome (Figure 3). Consequently, the depletion of JAZ proteins results in the transcriptional activation of transcription factors (TFs) and the subsequent expression of genes responsive to JA signaling pathways [81]. Amongst plant natural products, JAs play a key role in plant defense. JA biosynthesis depends on the circumstances in which the plant is grown. Genetic engineering can be used to change the genetics of JA synthesis to form more stress-resistant cultivars. Moreover, exogenous JAs are mostly used as anti-stress protective mediators by improving plant resistance. Further development of nano-biotechnology will ease the improvement of Nano carriers for efficient transfer of exogenous JAs to directed plant cells [82].

## 7. Role of Jasmonic Acid under Abiotic Stresses in Plants

The specific roles of JA under abiotic stresses of plants have been investigated broadly. JA has the capacity to enhance the tolerance of plants against a wide spectrum of abiotic stress factors, including, but not limited to, drought, salinity, high temperature, cold, and stress induced by heavy metals [74]. JA and its derivatives play a crucial role in a plant’s defense against both biotic and abiotic stresses. The functions performed by JAs in protection growth and mobilizing plant defense responses constitute a direct path for stress reduction. In response to abiotic stressors, JAs primarily enhance tolerance by activating the plant defense mechanisms, mostly characterized by the induction of antioxidant enzymes and other protective compounds (Figure 4) [78]. The use of jasmonates as a foliar application could help improve the plants’ capability to mitigate the effects of abiotic stresses in different plants. There is an enhanced negative impact of heavy metals when they are contaminated in the growth medium of soil for plants [12]. The plant showed decreased physiological growth which leads to stunned height and fruit falling at an early stage [83]. For example, applying the chromium (150 and 300 µM) in Choy sum (*Brassica parachinensis*) caused a deep reduction in the root and shoot length, biomass, and plant growth. It also showed leaf necrosis and death of plant cells. Moreover, it also altered the composition of the antioxidant enzymes like DHAR, HAR, GST, SOD, APX, and many others [51,83,84]. In this experiment, when we applied the JA at the foliar site’s quantity of 5, 10, and 15 µM, it promptly detoxified and neutralized the harmful effects of chromium (specifically at the highest doses). The recovery was noted as an improvement in photosynthetic pigments, gaseous exchange increase, and reduction in adsorption of chromium and stabilized minerals homeostasis as well [23,85].

In conclusion, it highlights the beneficial effects of jasmonic acid (JA) in enhancing plant tolerance to various abiotic stresses, encompassing drought, salinity, heat, cold, and heavy metal stress. However, there is a need for further experiments to fully understand JA’s involvement in mitigating abiotic.

### 7.1. Drought Stress

Drought stress is widely considered the primary abiotic stress, exerting a significant influence on agricultural productivity each year. Drought-induced stress disrupts the typical growth chronology of plants, leading to alterations in various biochemical, physical, physiological, and molecular pathways. Consequently, this adverse environmental condition culminates in diminished plant growth, ultimately leading to reduced agricultural productivity and lower crop yields [23,86,87]. Drought stress reduces the cell water potential and turgor pressure which results in the elevation of inter or intracellular solute concentrations. This elevated concentration causes a very harmful effect on plant growth [31,61,88,89]. There have been several reports on the role of JA in mitigating the adverse effects of drought stress on plant physiology but unfortunately, less information has been achieved from the experiments about the role of jasmonate against drought conditions. However, some studies showed that the level of JA increases in the roots and leaves of some plants during drought stress (Figure 5). In rice, the carboxyl methyl transferase gene (*AtJMT*) is depicted by the overexpression of a high level of JA release under drought stress [18,50,61]. This indicates that plants can produce more and more JA under drought stress. Collectively, JA production and signaling in plants under drought stress is the vital adaptation strategy of plants to cope with drought stress [8,23,90]. Using an exogenous spray of plant growth regulators, like JA, is considered one of the best techniques to overcome drought stress in plants [8,18].

There have been many reports that showed that an exogenous spray of methyl jasmonates reduces the effects of drought stress by activating more antioxidant enzyme activities [89]. It was reported by Alam et al. (2014) that an exogenous spray of JA under drought conditions enhanced antioxidant enzyme activity and decreased oxidative stress in the maize plants [91]. Furthermore, in other research, it was reported that JA significantly decreased the lipid peroxidation process within peanut seedlings subjected to drought-induced stress conditions by mitigating oxidative stress and enhancing antioxidant enzyme activities [51,92,93].

A study conducted by [91] demonstrated that foliar application of methyl jasmonate resulted in a significant enhancement of critical enzyme activities, including glutathione and catalase, within Brassica species. Thus, JA inhibited the reactive oxygen species (ROS) production under drought stress. It is not obvious how methyl jasmonates act to modify or alter the antioxidant mechanisms of Brassica or other plants. However, it might be possible that JA could modify gene transcriptions, translations, and post-transcriptional changes. However, JA is certainly responsible for changing the metabolic reactions at the cellular level in plants. Methyl jasmonate could also justify the water potential in plant cells. This all is interconnected with the regulation of JA production because it has a major role in drought stress coping in plants where it stimulates the plant metabolic machinery to produce secondary metabolites like different proteins, solutes, etc. One of these solutes is proline which has a key role in combating drought stress in plants [22,52,54]. Another study [94] reported that an exogenous spray of JA greatly improved the production of proline under drought conditions and this proline made barley plants capable of coping with drought conditions by producing more water potential and defensive proteins. Jasmonic acid also adjusted glutathione and ascorbate metabolism activities in *Agropyron cristatum* leaves under drought stress conditions [8,95]. There are also important roles of JA when it is applied exogenously or produced endogenously. When JA is produced it promptly stimulates the production of growth hormones or regulators. The exogenous foliar application of JA induces the synthesis of Abscisic acid (ABA) through a metabolic interconnection, thereby promoting ABA production in other research, it is observed that JA production could have an initiator effect on ethylene biosynthesis in many plants [51,96,97]. It suggests that with the production of JA, ethylene is also produced. So, JA has a synergistic effect like it is interlinked with the other growth hormones and regulators (Figure 5). In other research, it was found that by applying JA, the spermidine content was greatly increased in barley plants. The production of spermidine saved the cell membranes of barley from peroxidation because it has an activity like antioxidants. By applying a foliar spray of JA, the oxidative stress was alleviated and other harmful impacts were also increased because, due to the application of JA, the root growth was stunted and leaf and chlorophyll synthesis was also decreased in some plants [58,97,98]. The lack of leaf growth and chlorophyll contents in the plants is due to the epinastic effect of the JA. However, a foliar spray of JA degrades the chlorophyll and microtubules in some crops [23]. JA plays a pivotal role in mitigating the drastic impacts of drought stress which is very lethal in severe conditions.

In conclusion, JA production and signaling serve as vital adaptation strategies for plants to cope with drought conditions. Additionally, the exogenous application of methyl jasmonate can enhance antioxidant enzyme activities, reduce oxidative stress, and promote the production of crucial compounds like proline to provide an additional layer of defense against drought. 

### 7.2. Jasmonic Acid (JA) Robust Plant Tolerance to Salt Stress

Salinity is a big problem affecting plant growth and development. About more than 10% of the world’s arable lands are being affected by salinity and alkalization. Salinity and alkalization typically hinder the growth of plants and plants become unable to take up their nutrients from the soil [23]. These types of soils have very high contents of sodium and chloride ions resulting in decreased soil water potential. More than 20% of agricultural lands are being impacted by salt stress [18,99]. The main reason is that the electro-chemicals are mixed up with the irrigation water which creates problems of salinity because, when water is absorbed by the plants, the salt remains undissolved in the roots and shoots. Thus, this leads to the death of plants by accumulating too much salt in the plant cells [11,35,37,49]. Other factors include the discharge of toxic substances into the soil and the usage of inorganic fertilizers which severely increase salinity in root surface area. Plants absorb saline elements along with the uptake of nutrients from saline soil which causes stunted growth and poor nourishment [61,100]. Water and soil salinity create huge trouble in obtaining different nutrients, membrane permeability, metabolism of carbon sources, and nitrogen cycles. It also affects the chlorophyll biosynthesis, transpiration rate, and respiration rate as well [63,101,102]. In very severe conditions, the salinity causes the production of ROS in crop plants which results in potential oxidative stress resulting in the damage of DNA, peptides, proteins, and deactivating enzymes [50,100]. So, due to this, lipid peroxidation is also increased. JA as an osmotic regulatory agent/biochemical has vital importance in the biosynthesis system of plants because JA has the potential effects to cope with saline conditions in plants. Normally, during saline conditions, the endogenous JA production increases which has been demonstrated in the leaves of *Iris hexagona* [103] providing evidence that JA mitigates the dangerous effect of salt stress in almost all plants. In another study, it showed that JA boosted the biomass production of rice but inhibited the effect of salt which was present in water or soil [57,58,104]. In *Pisum sativum*, the JA application improved the profile of protein synthesis against salt stress [54,55]. Foliar application of JA showed a great improvement in soybean plants against salty water and soil because the soybean seedlings’ development was increased although there was salt stress. Foliar spray of JA under high salt stress showed improvement in the physical and physiological performances of safflower by improving the chlorophyll content, plant biomass, relative water content, and grain yield. In other research, the exogenous spray of JA also increased antioxidant enzyme activities, decreased lipid peroxidation, and enhanced the content of potassium in plants under salt stress [105,106].

In recent investigations, it has been reported that the application of JA via foliar spray resulted in a substantial increase in the proportion of total oil content, as well as elevated levels of linolenic and linoleic acids relative to oleic acid, particularly under conditions of salt-induced stress (Figure 5). This enhancement in fatty acid composition was attributed to the upregulation of lipoxygenase enzymatic activities [107,108]. The heightened lipoxygenase activity positively influenced the profile of soybean oil, augmenting the unsaturated index (UI) through the release of jasmonates. Additionally, another investigation demonstrated that the foliar application of JA led to the stimulation of H^+^-ATPase production and activity in the root system, resulting in reduced sodium uptake by soybean [109], promising JA as a mitigating agent for salt-induced stress in plants.

In conclusion, the application of jasmonic acid (JA) through foliar spray is presented as a potential solution to alleviate the negative impacts of salt-induced stress on plants, with evidence supporting its positive impact on biomass production, protein synthesis, antioxidant enzyme activities, oil content, and linolenic and linoleic acid content.

### 7.3. The Role of Jasmonic Acid under Heat Stress Conditions

High temperature (HT) is the eminent dilemma of the modern era because of the intense climatic variations that cause massive losses in the production of crop yield and ultimately endanger food security. The yield loss due to heat stress may thrust living beings to nutritional deficiencies [8,95]. Recently, a research study conducted in African tropical and subtropical zones on maize crops revealed a yield loss of approximately 1–2%, respectively, when a single-degree elevation in temperature occurred beyond 30 °C [99,110]. Under heat stress, many pathways are activated by plant metabolism to mitigate the effect of heat stress. These all are interlinked with gene expression by which heat shock proteins are produced as special proteins of low molecular weight. The principal mechanism employed by plants to mitigate the effects of heat stress involves the synthesis of heat shock proteins. This mechanism constitutes a conserved molecular system in many plants to produce heat shock proteins to mitigate the effects of high temperatures. JA is recognized as a prominent phytohormone with a pivotal role in the biosynthesis of numerous secondary metabolites, as well as the induction of heat shock proteins and other phytohormones. Consequently, JA assumes a significant regulatory role in the modulation of heat shock protein production within plants [95,111].

In Arabidopsis, the distinct involvement of JA signaling has been observed in the process of synthesizing heat shock proteins. According to research, it has been concluded that enhancing the level of JA under heat stress is considered a defense activity of plants to produce heat shock proteins under heat stress [23,51]. The foliar spray of JA in lower doses showed the preservation of cell structure under heat stress as obvious by leaf electrolyte leakage assays. In other research conducted on barley, it was revealed that a foliar spray of JA on the barley plants improved the signaling of gene expression and heat shock proteins as a response against heat stress [90,109]. Some other studies reported that the foliar application of JA modified the phenolic profiles of compounds at the cellular level in plants which resulted in the production of many heat shock proteins. These proteins make plants very resistant to heat stress. Moreover, elevating abscisic acid content in cells of plants along with JA might be very helpful in decreasing the harmful effects of heat stress as it induces the production of more heat shock proteins [18,97]. However, Abscisic acid also regulates the closing and opening of stomata and water-saving strategies under heat stress. There is a need for more research to fully understand the effects of heat stress and the role of JA under heat stress in plants [95,109]. Many phytohormones mainly control heat stress. However, JA is one of them because its production in plants mainly reduces heat stress. 

In conclusion, JA has the potential to increase plant defense mechanisms against heat stress, including gene expression and phenolic compound profiles. However, further research is imperative to understand the impacts of heat stress and the involvement of JA in enhancing plant defense. A detailed illustration of heat stress and production of JA and ethylene under heat stress in plants is given below in Figure 6.

### 7.4. Jasmonic Acid Enhances Plant Tolerance against Cold Stress

Low temperature is also considered very lethal for the growth and nourishment of plants. Plant growth is optimum during an optimal growth temperature. When the temperature deviates from the optimal range, plants typically undergo a series of physical, biochemical, physiological, and molecular alterations as part of their normal response [112]. Plants have evolved such a mechanism to combat growth rate under different environmental conditions to maximize their growth by producing defensive system signals that allow fast reproduction of the plant cells. This mechanism is called molecular homeostasis which performs its functions of growth regulation under different temperatures. Typically, plants suffer from cold or chilling stress between the 0 and 15 °C range of temperature. They usually mitigate cold stress by modifying their gene expression, and physical, biochemical, and physiological modifications in their biosynthetic systems [59,78,113]. During the last few decades, there has been a lot of attention on understanding the roles of JA as a signaling molecule towards cold or chilling stress. During different research, it has been found that numerous phytohormones like JA, Abscisic acid, and ethylene have a great role in plant physiology to alleviate the effect of low-temperature stress or chilling stress. In research, it was learned that the JA content was drastically increased during the low-temperature stress (Figure 6). Moreover, the JA content was also increased in the leaves of *Pinus pinaster* when they were highly exposed to chilling stress [20,50,107]. Similarly, using methyl jasmonate treatment before harvest enhanced lemons’ resistance to postharvest chilling damage and deterioration without compromising production and quality [114].

In different research, it has been studied that JA stops the effect of chilling injuries to plant cells by different protein inhibitors, polyamines, and cryoprotective agents and upsurges the activities of LOXs and antioxidants.

In this specific investigation, it was observed that exogenous methyl jasmonate (MeJA) augmented the activity of the ascorbate–glutathione (AsA–GSH) cycle, consequently enhancing the antioxidant capacity of root tissues. Simultaneously, the pivotal components of the antioxidant enzyme system, namely superoxide dismutase (SOD), peroxidase (POD), catalase (CAT), and the aforementioned ascorbate–glutathione (AsA–GSH) cycle, collectively played significant roles in scavenging reactive oxygen species (ROS) [115]. 

Upon exposure to cold conditions, the induction of jasmonic acid (JA) biosynthesis genes, specifically *OsLOX2*, is noted in plants, leading to a rapid increase in endogenous JA levels in rice [116]. Conversely, Arabidopsis mutants with inhibited JA production and signaling, specifically *lox2*-mutants, show sensitivity to freezing stress [117]. In the context of banana plants, the application of methyl jasmonate induces the transcription of MaLOXs, thereby enhancing resistance to low temperatures [118]. Furthermore, the supplementation of exogenous MeJA under low-temperature conditions amplifies the concentration of JA, subsequently improving the transcriptional and functional aspects of lipoxygenase (LOX) [119]. LOXs play a pivotal role in catalyzing the oxidation of polyunsaturated fatty acids, and their expression is intricately regulated by phytohormones [120]. Research on rice seedlings was conducted and showed that if we had applied the JA to seedlings before chilling stress, it would mitigate the effects of chilling stress at a later stage by osmoregulation of different antioxidants. Application of JA in the seedlings preserved the stomatal shape by enhancing hydrolytic activities [18,99]. JA has a strong role in coping with cold stress. It reduces the physiological and biochemical effects of cold stress.

In conclusion, molecular homeostasis assumes a significant function in regulating plant growth when subjected to fluctuating temperatures, with a focus on cold or chilling stress between 0 and 15 °C. JA emerges as a crucial signaling molecule in mitigating the effects of low-temperature stress, influencing gene expression, antioxidant activity, and stomatal regulation.

### 7.5. Jasmonic Acid Signaling Helps Plant to Tackle Flooding Stress

Naturally, water is a fundamental requisite for plant growth; however, an excess of water can induce deleterious impacts on plant growth. Plants have adapted various strategies to cope with the flooding situation like adventitious root production, aerenchyma production, root hair development, and shoot growth to mitigate the effect of flooding. We can define flooding as a situation for plants when only plant roots are exposed to a very high content of water and this excessive flooding water causes hindrances in the gas transportation in the rhizosphere portion of plants [96,108]. Usually, oxygen dispersion is 10,000 times slower in water as compared to the atmosphere. So, water logging conditions may be a severe disadvantage for plant growth. Jasmonic acid has been found to play a very vital role in the production of aerenchyma and root hair development because it, along with ethylene, has a better impact on plant growth under flooding situations. In another investigation, the role of ethylene in the formation of aerenchyma was shown. So, ethylene along with JA plays a significant function in producing root hairs to alleviate the effect of flooding stress [23,83].

The exogenous application of JA through foliar spray enhances the production of ethylene. Thus, JA may increase the production of ethylene which is very helpful for aerenchyma cells. In soybeans, it was observed that foliar spray greatly increased the ethylene contents which led to aerenchyma cell development and, overall, this reduced the effects of flooding water [35,47]. Flooding stress is controlled by different phytohormones, and JA is one of them. Taking these mechanistic strategies into consideration, the possible mechanisms under flooding stress are portrayed (Figure 7).

### 7.6. Mitigation of Heavy Metals Stress through JA

Modernization and industrialization have increased the number of environmental pollution issues [49]. Among them, heavy metals (HMs) play a crucial role in ecological deterioration, posing a severe threat to plants and other living organisms [11]. HMs are not biodegradable, exhibit prolonged environmental persistence, and endanger human health if they get into the food supply [38]. While numerous HMs, like Cd, Pb, and Hg, are not necessary for plant growth, others including Fe, Mn, Cu, Mo, and Zn are required micronutrients for plant growth but have adverse effects on plant development at high concentrations [34,121]. In another study [57,122], heavy metals are considered highly poisonous for the growth of plants because these metals/metalloids are present in the environment and are highly absorbable when present in soil. These elements combine with water molecules and penetrate into root cells via micropores in root hairs. Several experimental studies have been conducted on the interaction of heavy metal stress with different hormones like salicylic acid, auxins, jasmonic acid, and some others. Heavy metals are very toxic when absorbed by the vascular tissues of plants as these reduce the oxidation-reduction reactions, ATP synthesis, chlorophyll activity, and cell division (Figure 7). Hence, plants move towards stunned growth as well as cell death [13,17,34]. Here, we are going to address the relationship of jasmonic acid (JA) with heavy metals where JA acts as a repellent to the toxicity induced by heavy metals and it promptly boosts up the immunity system in plants by expressing defense-related genes [23,36,47,122]. Additionally, ascorbic acid and glutathione molecules play crucial roles in the plant defense system and function as cofactors for enzymes, impacting plants’ growth and development. Non-enzymatic antioxidants also comprise these substances [35]. JA, along with its methyl ester derivative known as methyl jasmonate (MJ), constitutes a class of oxylipins. These compounds are synthesized endogenously within numerous higher plant species via the enzymatic oxidation of fatty acids mediated by lipoxygenases [36,113]. The interaction between JA/MJ exerts a pronounced influence on a diverse array of physiological and biochemical mechanisms, alongside its pivotal role in regulating plant growth and development. JA/MJ generally prevents the opening of stomata, cell division, photosynthetic processes, the creation of flower buds, seed germination, and embryogenesis [35,123]. Similarly, JA/MJ also promotes the creation of tubers, the senescence of leaves, the abscission of petioles, the ripening of fruit, the degradation of chlorophyll, the synthesis of carotenes, and proteins [37,55]. Additionally, in climacteric apple fruits, the application of methyl jasmonate (MJ) enhances the synthesis of ethylene, esters, alcohols, and acetic acid [47]. The matrix shows that specific pressures might have favorable and unfavorable impacts on plants. Hence, the utilization of emerging phytohormones such as jasmonate appears promising for mitigating heavy metal-induced stress in plants and enhancing their physiological and biochemical attributes affected by metal stressors. These phytohormones hold the potential to foster the development of plants endowed with greater tolerance against the challenges posed by heavy metal stresses. Methyl jasmonate is a member of the cyclopentanone chemical family and is an organic plant hormone that engages in signal transduction pathways in the response of plants to diverse environmental stressors [7]. Pathogenic infections, wounds, ozone, and conditions associated with heavy metals, all increase endogenous methyl jasmonate levels [18]. Me-JA applied exogenously or as a pretreatment showed multi-stress resistance and can decrease Cd damage by reducing metal absorption [52].

JA plays a crucial role in various plant species to tackle the detrimental effects caused by heavy metal stress [50]. The application of JA has reduced the metal-induced stress in pigeon peas [57,122]. The elevation in endogenous JA levels is directly proportional to the effects caused by heavy metal stress. An experimental study on *Arabidopsis thaliana* showed that JA levels increased by four and six times in 7 h of treatment with Cu (100 µM) and Cd (100 µM), respectively [38,49]. Similarly, a research study on peeper and rice showed that the cellular level of JA increased two times more in pepper when it was subjected to 50 mg/L Cd for 48 h. Likewise, the level of JA increased by nine fold in rice when treated with 100 mM Cu for 6 h. These findings show that heavy metal concentrations are directly proportional to JA levels [86]. During these investigations, it was obvious that when pepper plants were treated with Cd, the release of jasmonic acid had a positive impact on plant roots, chlorophyll, cell division, and cell growth and reduced the toxic effect of cadmium on pepper plants when exposed to the exogenous environment. In another experiment conducted, it was found that lead (pb) reduced the chlorophyll content and carotenes in the *Wolffia arrizha* plants but when they were treated with exogenous JA, it promptly increased the level of immunity, and the plant escaped from the effects of pb-induced toxicity by reducing the adsorption of pb into chlorophylls and carotenoids [18,83,122]. Collectively, these investigations suggest that JA has a very positive correlation with plant growth against heavy metal stress. By adopting modern genetic approaches, the role and mechanism of JA are completely demonstrated in different research. In Arabidopsis plants, it was observed that the AOS mutant, which exhibits genetic mutations in the JA biosynthetic pathway, displayed a greater sensitivity to cadmium (Cd) exposure. Notably, the AOS mutant exhibited a more pronounced cellular accumulation of Cd and manifested a more severe albino phenotype induced by Cd exposure in comparison to the wild-type plant [36,58]. Considering Arabidopsis as a model plant, it was found that selenium decreased the root and shoot growth and the mutated *JAR1-1* mutant effect by selenium was more astonishing as compared to the wild type. The application of JA was also observed in tomato *SPR2* mutants which lacked jasmonic acid genes [24,34]. When the *SPR2* and wild type were subjected to a similar level of cadmium, the *SPR2* mutant showed a very severe phenotype compared to the wild type with reduced growth and nourishment. Compared to JA signaling mutants, the transgenic plants have a highly overexpressing level of jasmonic acid biosynthetic genes with the increased level of JA and showed great resistance against heavy metal toxicity [124]. As an example, *Gossypium hirsutum GHAOS* activated JA biosynthesis and showed improvement in the stress of the heavy metal. Moreover, the rate of sustainability of *GHAOS* overexpression of transgenic plants under the treatment of copper conditions was two-fold more compared to the wild-type plants (Figure 7). Similarly, wheat genes like *TaAOS*, and its overexpression ratio, increased the plant’s tolerance to withstand the adverse effects of heavy metal stress caused by zinc. Furthermore, in tobacco plants the over-expression of *TaAOs* was investigated and three times more JA level was observed compared to the wild type. These investigations make the mitigative role of JA crystal clear against heavy metal toxicity. The level of JA can be changed or enhanced with the alteration of some relevant genes by genetic modifications. In grapeseed genotypes, the gaseous exchange attributes were negatively affected by Cd stress whereas, the exogenous application of JA increased cellular JA level which ultimately improved the gaseous exchange and resulted in elevated transpiration rate and energy production in the form of ATP [7,37,50].

In conclusion, JA possesses the potential to assume a crucial role in alleviating the deleterious impacts of HM toxicity on plants. However, further research into the specific pathways and underlying molecular mechanisms implicated in the alleviation of HM stress by JA would provide a more comprehensive understanding.

## 8. Conclusions and Future Perspectives

Plants have evolved special phytohormones, such as jasmonic acid (JA), that confer tolerance primarily by activating plant defense mechanisms, especially antioxidant enzymes. Further, the interactive mode of JA with these enzymes protects plants against the toxic effects caused by abiotic as well as biotic factors. Exogenous JA is regarded as a bio-stimulant that induces the expression of JA-related genes under drought, salinity, flooding, and heavy metal stress environments. Genetic modification is also a possible approach to a substantial upsurge in the production of JA leading to an enhanced resistance mechanism in plants. Recent advances in transcriptomics, genomics, and proteomics have unraveled the interaction of gene and protein networks. These techniques have also unfolded the synergistic crosstalk of JA with other phytohormones such as ABA, ET, and SA, etc. However, the behavior of phytohormones and their signaling networks are unpredictable and respond differently to different stress factors. Therefore, future studies should focus on the molecular mechanism of JA involvement and their synergistic or antagonistic interaction with other phytohormones, their signaling pathways, and biosynthesis under different environmental conditions to understand the underlying mechanisms. This will help scientists breed climate-smart plants that can improve crop yield and production under the harsh environmental conditions caused by climate change.

## Figures and Tables

**Figure 1 plants-12-03982-f001:**
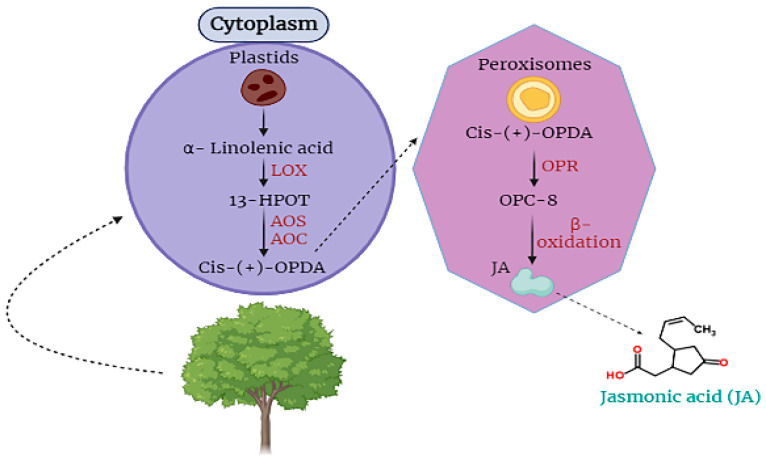
Biosynthesis and translocation of jasmonic acid (JA) in plant cells.

**Figure 2 plants-12-03982-f002:**
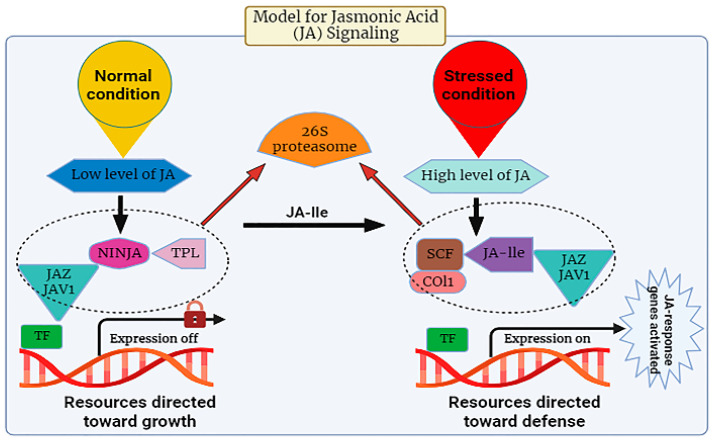
The expression levels of jasmonic acid under normal and stressed conditions.

**Figure 3 plants-12-03982-f003:**
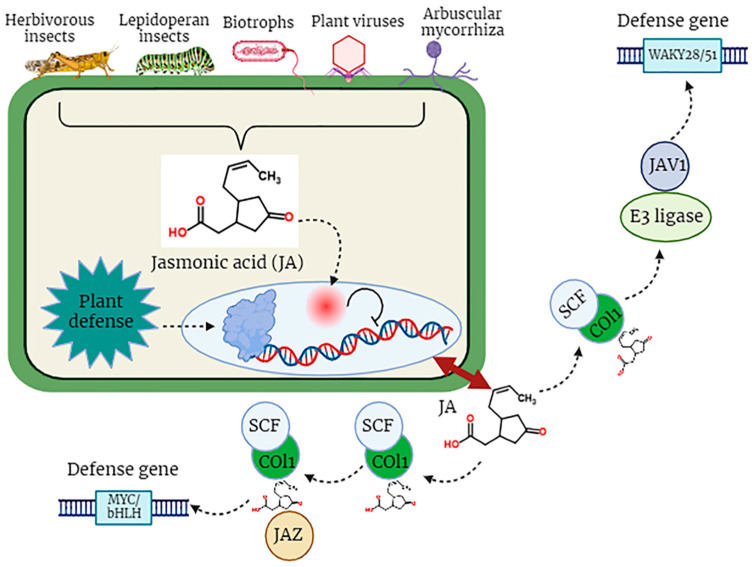
The defensive role of jasmonic acid (JA) in plants against biotic stress factors.

**Figure 4 plants-12-03982-f004:**
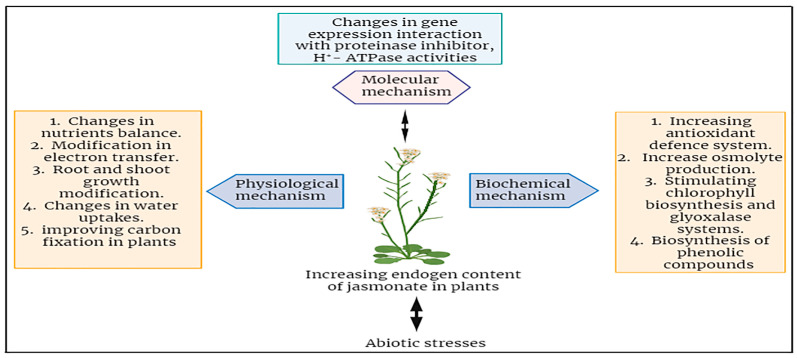
Summary of jasmonates mechanisms in enhancing abiotic stress tolerance in plants.

**Figure 5 plants-12-03982-f005:**
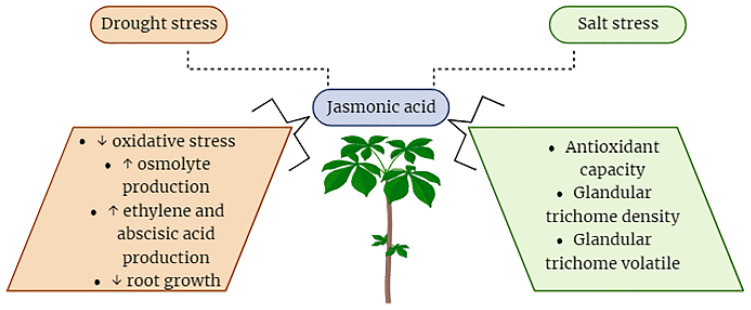
Belligerent role of JA in plant defense under drought and salinity stresses. Arrows in the figure indicate up and down production levels.

**Figure 6 plants-12-03982-f006:**
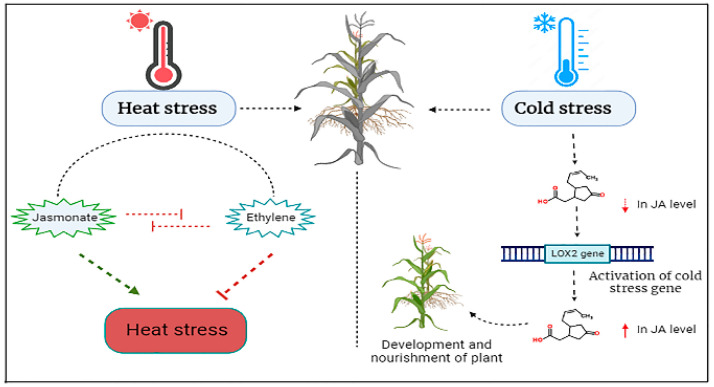
Schematic figure of JA and ethylene production under heat stress in plants and Switching on the genes of cold stress and production of JA under cold stress. Arrows in the figure show different levels of JA or stresses.

**Figure 7 plants-12-03982-f007:**
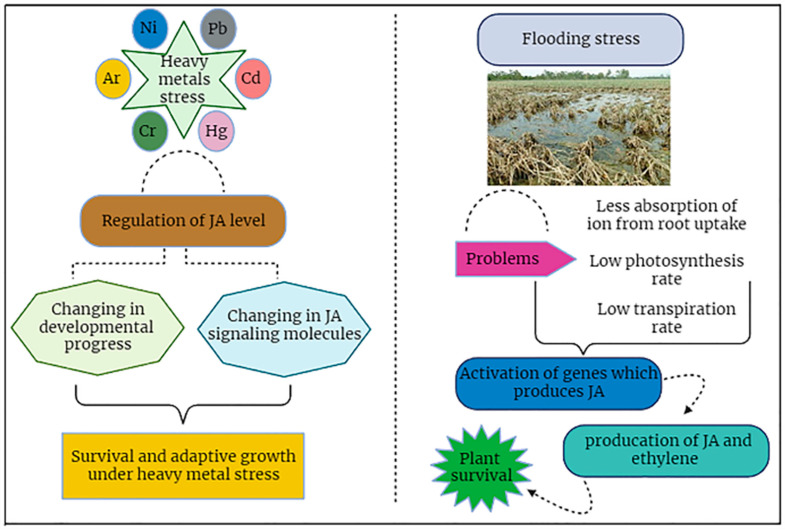
JA signaling plays a pivotal role in plant survival under heavy metals and flooding stress conditions.

**Table 1 plants-12-03982-t001:** Some physiological responses in plants are elicited by the activation of JA-responsive genes.

Plant Species	Gene	Response	Ref:
*Arabidopsis thaliana*	*DAD1*	The activation of this gene leads to the synthesis of phospholipase A1, resulting in reduced filament elongation and a delayed anther dehiscence.	[65]
*Oryza sativa*	*chi11 & ap24*	Sheath Blight tolerance	[66]
*Arabidopsis thaliana*	*RSI1 & RRTF1*	These genes constitute a functional module dedicated to the preservation of the infection memory	[67]
*Arabidopsis thaliana*	*NPR1*	This gene is involved in both SA and JA signaling pathways in *A. thaliana* defense	[68]
*Zea mays*	*WRKY*	These genes are involved in plant defenses against herbivore attack	[69]
*Nicotiana tabacum*	*JERF1*	Gene involved in defense against salt stresses	[70]
*Oryza sativa*	*AtJMT*	It causes over-expression of the production of ABA acid and Jasmonates in panicles	[71]
*Nicotiana tabacum*	*Pvgstu*	Gene responsible for resistance to combined heat and drought	[72]
*Arabidopsis thaliana*	*ORA59*	This gene has a strong role against biotic stresses	[73]

## Data Availability

Data are contained within the article.

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
