# Peer review of "The Multifaceted Role of Jasmonic Acid in Plant Stress Mitigation: An Overview"

_plants, 2023, doi:10.3390/plants12233982_

Round 1
Reviewer 1 Report
Comments and Suggestions for Authors
The present manuscript, entitled "The Multifaceted Role of Jasmonic Acid in Plant Stress Mitigation: An Overview," addresses a highly relevant topic for plant physiologists. I accepted the opportunity to review this paper with the aim of gaining a deeper understanding of JA metabolism and its functions, and I found that this paper met my expectations. I learned a great deal from reading it. The manuscript provides a clear description of various significant aspects of JA's actions, covering both molecular and physiological perspectives, all presented in an engaging manner. Only a minor correction on line 262 is needed. I would like to extend my congratulations to the authors for this comprehensive review.
Author Response
We want to express our gratitude for your positive feedback.

Reviewer 2 Report
Comments and Suggestions for Authors
According to the study of the manuscript with the number “plants-2706832” and tittle “The multifaceted role of Jasmonic acid in plant stress mitigation: An Overview” I did not find any major loopholes. The authors have nicely covered all the aspects related to the role of Jasmonic acid in biotic and abiotic stress, and their graphical presentation is also much appreciated. The only lacking point in this review is the cross talk of JA with other hormones, but as the authors have already mentioned it in the conclusion and future perspective section that they will cover this point in their future research so the manuscript can be accepted in its present form.
Author Response
We greatly appreciate your feedback and valuable suggestions. We will certainly take your thoughtful suggestion into account in our upcoming research endeavors.

Reviewer 3 Report
Comments and Suggestions for Authors
The article aimed to present an overview of the role of Jasmonic acid in plant stress mitigation. Generally, it is a good review that deserves to be published. There are some suggestions:
Group similar abiotic factors together to improve the flow. For example, mention all types of stress (salinity, heat, drought, freezing, water logging) together and then discuss heavy metal and other micropollutants as, for example, drugs stress separately.
Provide more detail on how phytohormones regulate normal plant growth and maintain homeostasis against abiotic stress. Consider offering examples or specific mechanisms.
Use consistent terminology. For example, if you refer to "phytohormones" at the beginning, continue using that term consistently throughout the text.
Expand on the functions of JA in plant cells, providing more information about how it influences cell division, reproductive growth, fruit ripening, and other physiological processes.
Provide specific examples of how phytohormones, including JA, interact with plant physiology to alleviate the effects of low-temperature stress. Mention concrete findings from the research studies.
Connect the concept of JA as a signaling molecule with its role in mitigating the effects of low-temperature stress. Explain how JA interacts with protein inhibitors, polyamines, cryoprotective agents, LOXs, and antioxidants.
I suggest adding a short chapter about the role of JA in micropollutant stress.
Comments on the Quality of English LanguageThe manuscript requires some minor English language copyediting
Author Response
We sincerely appreciate your thorough review of the manuscript and your valuable suggestions. We have made the necessary revisions to address the queries in the respective sections.
